# CyclePhase: Robust phase detection in cardiovascular imaging through cyclic motion estimation

**Soufiane Ben Haddou**[*1,2]                   S.BENHADDOU@AMSTERDAMUMC.NL
**Rudolf L. M. van Herten**[*1,3,4]                RLV4001@MED.CORNELL.EDU
**Connie R. Bezzina**[5]                       C.R.BEZZINA@AMSTERDAMUMC.NL
**R. Nils Planken**[6,7]                        PLANKEN.NILS@MAYO.EDU
**Joost Daemen**[8]                           J.DAEMEN@ERASMUSMC.NL
**Jolanda J. Wentzel**[8]                       J.WENTZEL@ERASMUSMC.NL
**José P. Henriques**[9]                        J.P.HENRIQUES@AMSTERDAMUMC.NL
**Ivana Išgum**[1,6,7,9]                         ISGUM.IVANA@MAYO.EDU

[1] *Department of Biomedical Engineering and Physics, Amsterdam UMC, University of Amsterdam*

[2] *Informatics Institute, University of Amsterdam, Amsterdam, The Netherlands*

[3] *Department of Radiology, Weill Cornell Medicine, New York, United States of America*

[4] *Cornell Tech, New York, United States of America*

[5] *Department of Experimental Cardiology, Amsterdam Cardiovascular Sciences, Heart Failure & Arrhythmias, Amsterdam UMC, The Netherlands*

[6] *Department of Radiology and Nuclear Medicine, Amsterdam UMC, University of Amsterdam*

[7] *Department of Radiology, Mayo Clinic, Rochester, United States of America*

[8] *Department of Cardiology, Erasmus University Medical Center, Rotterdam, The Netherlands*

[9] *Amsterdam Cardiovascular Sciences, Amsterdam, The Netherlands*

**Editors:** Accepted for publication at MIDL 2026

## Abstract

Accurate cardiac phase detection is essential for cardiovascular imaging applications requiring temporally aligned measurements. While existing methods treat phase detection as discrete frame classification, we propose a fundamentally different approach that models cardiac phase as a continuous cyclic variable on the unit circle. Our method introduces gradient-based input transformations to isolate motion from static anatomy, thereby making it robust to appearance variations, such as calcifications, in intravascular ultrasound (IVUS). Through multi-objective optimization combining temporal consistency via Earth mover's distance with continuous phase regression, we achieve superior performance across both IVUS and cardiac MRI. Experiments demonstrate that explicitly modelling cardiac periodicity yields more accurate and temporally coherent phase detection compared to classification-based approaches, with particular improvements in artefact-heavy clinical scenarios. Our unified framework eliminates the need for modality-specific preprocessing or segmentation masks, providing an end-to-end solution for cardiac motion characterization.

**Keywords:** deep learning, periodic motion estimation, intravascular ultrasound, cine MRI

---

* Contributed equally

## 1. Introduction

Accurate cardiac phase detection is critical for cardiovascular imaging applications that rely on consistent and temporally aligned measurements. Cardiac motion exhibits semi-periodic behaviour, and variations in heart rate introduce temporal inconsistencies across cycles. Therefore, extracting phase-aligned frames is essential to obtain reliable and comparable measurements to observe cardiac function and detect pathology without temporal interference, or standardized temporal alignment for tasks that utilize all frames. This fundamental need for a standardized way to describe motion present in the image makes phase detection essential for applications such as cardiac and coronary motion quantification (Lyu et al., 2021; Bajaj et al., 2021), image registration (Alvarez-Florez et al., 2023; Garzia et al., 2024), and standardized downstream analyses (Chen et al., 2024).

Despite this shared need across cardiovascular imaging, current approaches to cardiac phase detection have evolved in isolation within different modalities. In intravascular ultrasound (IVUS), catheter pullback motion is addressed through gating strategies that identify end-diastolic (ED) frames to recover smoothly varying arterial images (Shi et al., 2017; Bajaj et al., 2021; Sun et al., 2024). In cine MRI (CMR), existing methods similarly require pre-identification of specific cardiac phases (Alvarez-Florez et al., 2023) or rely on segmentation masks as prerequisites to identify ED and ES frames (Chen et al., 2024) or to structure temporal data. This modality-specific development has resulted in fragmented solutions that, while sharing the common goal of characterizing cardiac motion, remain siloed and dependent on intermediate processing steps. Moreover, by treating phase detection as discrete frame classification rather than modeling the continuous periodicity of the cardiac cycle, these methods often fail when image appearance alone becomes unreliable—such as in the presence of calcifications in IVUS (Sun et al., 2024) or other out-of-distribution anatomical features.

We present a fundamentally different approach that treats cardiac phase as a continuous cyclic variable rather than discrete labels. By formulating phase detection as regression on the unit circle, we naturally incorporate the periodic nature of cardiac motion into both representation and learning, avoiding temporal discontinuities that hinder classification-based methods. Our framework introduces gradient-based input transforms to isolate motion from static anatomy, making the method robust to appearance variations like calcifications. Through a multi-objective optimization combining temporal consistency via Earth mover's distance with continuous phase estimation, we achieve superior performance across both IVUS and CMR modalities. This demonstrates that explicitly modeling cardiac periodicity—rather than treating it as a byproduct of frame classification—yields more accurate and temporally coherent phase detection in challenging clinical scenarios.

Building on the core insight that cardiac phase should be treated as a continuous cyclic variable rather than discrete labels, we develop a unified framework applicable across cardiovascular imaging modalities. Our approach addresses three key challenges identified in existing methods: (1) the need to model continuous periodicity, (2) the requirement for robustness to appearance variations, and (3) the importance of temporal coherence. To achieve this, we design a method that combines motion-focused preprocessing, circular phase representation, and multi-objective optimization. Our code is made publicly available on GitHub.

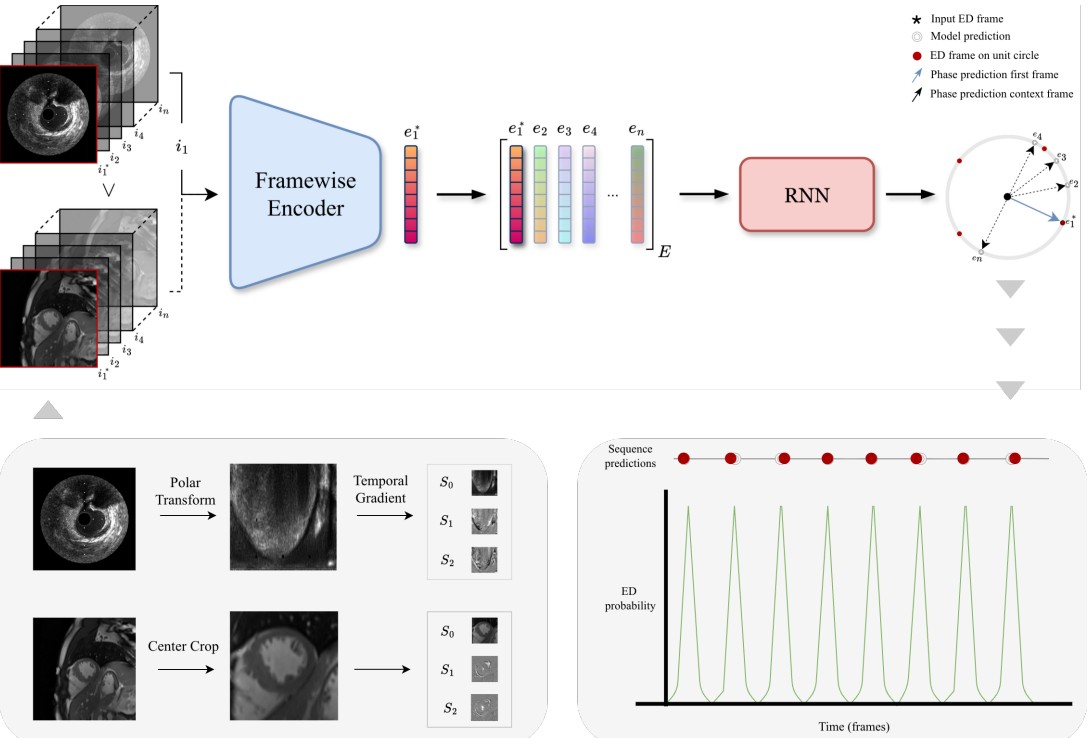

Figure 1: Overview of the proposed cardiac phase detection framework. Top: A sequence of IVUS or ($\lor$) cine MRI frames is processed by a framewise convolutional encoder, producing per-frame embeddings $e_1, \ldots, e_n$ that are passed to a temporal RNN. The RNN outputs both binary end-diastolic (ED) logits and continuous phase vectors on the unit circle, enabling joint ED frame detection and smooth phase estimation. Bottom left: Modality-specific preprocessing pipelines showing polar transformation for IVUS and center cropping for CMR, followed by temporal gradient extraction yielding state encodings $S_0, S_1,$ and $S_2$. Bottom right: Example output showing predicted ED events and corresponding ED probability trace over time.

## 2. Method

Given an image sequence $\{x_t\}_{t=1}^{T}$ with $x_t \in \mathbb{R}^{C \times H \times W}$, each channel corresponds to one of the gradient-based input representations described in Section 2.1, where setting $C = 1$ reduces to the case of a single raw image channel. The model produces per-frame event logits $\ell_t \in \mathbb{R}$ and a phase vector $\hat{\phi}_t \in \mathbb{R}^2$ parameterized on the unit circle. The following sections describe how each component contributes to robust phase detection.

### 2.1. Input transformations

**IVUS images:** In IVUS imaging, the vessel wall exhibits approximately circular geometry around the catheter, which is inherently positioned at the image center due to the

acquisition geometry. By transforming from Cartesian to polar coordinates, we convert the vessel's radial expansion and contraction into a simpler vertical displacement pattern that convolutional networks can easily capture. Specifically, we map Cartesian coordinates $\mathbf{u} = (x, y)$ to polar coordinates $\mathbf{v} = (r, \theta)$ using the image center as the reference point $\mathbf{c}$:

$$r = \|\mathbf{u} - \mathbf{c}\|_2, \quad \theta = \mathrm{atan2}(y - c_y, x - c_x). \tag{1}$$

The resulting rectified image $I^{\mathrm{pol}}(r, \theta)$ is sampled on a fixed grid where cardiac motion manifests primarily as vertical displacement, aligning well with the inductive biases of standard convolutional kernels.

**CMR images:** Traditional approaches compute the transformation center from myocardium segmentation masks (Rajiah et al., 2022), which creates an undesirable dependency on auxiliary annotations. To maintain our method's independence from intermediate processing steps and prevent potential label leakage, we process CMR frames in their native Cartesian coordinates. This design choice ensures our phase estimator remains end-to-end without hidden dependencies.

**Gradient transform:** To explicitly guide the model toward motion-relevant features, we augment the input with gradient information that highlights temporal changes. We compute gradients using Sobel filters in the $dt$ direction, creating a motion-sensitive representation that emphasizes edges and boundaries where cardiac motion is most apparent.

### 2.2. Continuous phase representation

The periodic nature of cardiac motion must be considered in both label design and loss formulation. Rather than treating phase as a linear variable that suffers from discontinuities at cycle boundaries, we embed phase values on the unit circle, naturally encoding periodicity into the representation itself.

Given end-diastolic (ED) anchor frames $\{\tau_k\}$ in ascending order, we assign each frame a continuous phase $\phi_t \in [0, 2\pi)$ that increases linearly within each cardiac cycle and wraps at $2\pi$. For frames between consecutive anchors:

$$\phi_t \;=\; 2\pi \, \frac{t - \tau_k}{\tau_{k+1} - \tau_k} \quad \text{for } t \in [\tau_k, \tau_{k+1}), \tag{2}$$

with $\phi_{\tau_k} = 0$, for all $k$. Edge segments before the first and after the last anchor receive uniform phase assignments consistent with this per-interval rule. Instead of regressing $\phi_t$ directly, we map phases to unit circle embeddings:

$$\boldsymbol{\phi}_t \;=\; \big(\cos \phi_t, \, \sin \phi_t\big), \qquad \|\boldsymbol{\phi}_t\|_2 = 1, \tag{3}$$

which avoids wrap-around discontinuities and ensures smooth transitions across cycle boundaries. Thus, it remains robust to heart-rate variability, as the circular distance between any two phases remains well-defined and continuous.

### 2.3. Network architecture

The model architecture follows an encoder-decoder architecture, where a framewise convolutional encoder is followed by a temporal recurrent neural network (RNN) decoder that produces phase predictions.

**Framewise encoder:** Each frame $x_t$ is independently processed by a 2D ResNet encoder $E_\theta$ (He et al., 2016), where a series of $S$ skip-connection blocks progressively reduces the spatial dimensions of the input by a factor of $2^S$ while preserving the full temporal resolution. A detailed description of our image encoder architecture is provided in Appendix A:

$$\mathbf{z}_t = E_\theta(x_t) \in \mathbb{R}^{C_e}, \qquad (H, W) \mapsto (H/2^S, W/2^S).$$

**Temporal decoder:** The encoded sequence $\{\mathbf{z}_t\}_{t=1}^T$ are fed into a recurrent neural network (GRU or LSTM) that models temporal dependencies:

$$\mathbf{h}_t = \text{RNN}_\psi(\mathbf{z}_t, \mathbf{h}_{t-1}) \quad (\text{or BiRNN with } \mathbf{h}_t = [\overrightarrow{\mathbf{h}}_t; \overleftarrow{\mathbf{h}}_t]).$$

A sequential prediction head operates on the hidden state $\mathbf{h}_t$:

$$\hat{\boldsymbol{\phi}}_t = \mathbf{w}_1^\top \mathbf{h}_t \in \mathbb{R}^2, \qquad \hat{\boldsymbol{\phi}}_t \leftarrow \frac{\hat{\boldsymbol{\phi}}_t}{\|\hat{\boldsymbol{\phi}}_t\|_2},$$

$$\ell_t = \mathbf{w}_2^\top \hat{\boldsymbol{\phi}}_t + b \quad \Rightarrow \quad \hat{y}_t = \sigma(\ell_t),$$

where $\ell_t$ produces logits for ED frame identification and $\hat{\phi}_t$ yields 2D vectors normalized to the unit circle $\mathbb{S}^1$. This allows joint optimization of discrete event detection with an enforced cyclic embedding through the continuous phase estimator $\hat{\phi}_t$, leveraging their complementary nature.

In addition to the GRU/LSTM decoder, we evaluate a Transformer-based temporal decoder as an alternative backbone in our ablations. We replace the RNN with a stack of Transformer encoder layers operating on the per-frame embeddings $\{\mathbf{z}_t\}_{t=1}^T$ (with Fourier positional embeddings), while keeping the same output heads for ED logits and unit-circle phase vectors. This allows isolating the effect of the temporal modeling choice under an otherwise identical pipeline.

### 2.4. Training objectives

Let $y_t \in \{0,1\}$ be the ED indicator and $\phi_t$ the phase target. We combine classification, temporal shape regularization, and circular regression in a single objective, echoing the clear multi-term objective presentation style used in the reference paper.

**Balanced binary cross-entropy:** End-diastolic frames are rare compared to non-ED frames, which makes a standard BCE dominated by the negative class. To ensure that both ED and non-ED frames contribute equally, we compute the binary cross-entropy on the two subsets separately: one loss over the frames with $y_t = 1$ and one over the frames with $y_t = 0$. We then average the available terms, so that each class has equal influence whenever it appears in the sequence:

$$\mathcal{L}_{\text{BCE}} = \tfrac{1}{2}\Big(\text{BCE}\big(\ell_t,\, y_t{=}1\big) + \text{BCE}\big(\ell_t,\, y_t{=}0\big)\Big). \tag{4}$$

This mirrors our implementation, where logits and labels are masked by class and the two BCE terms are averaged, yielding a balanced contribution from positive and negative samples.

**Temporal Earth-mover's distance (EMD):** Recall that $p_t = \sigma(\ell_t)$ denotes the predicted probability of an ED event at frame $t$, and $y_t$ the corresponding binary label. To encourage the model to place each predicted peak not only at the correct location but also with the correct temporal shape, we compare the cumulative distributions of the predictions and the labels within each sequence. Aligning these cumulative curves penalizes predictions that are shifted, spread out, or fragmented over time, and therefore promotes sharp, well-localized peaks that follow the temporal ordering of the cardiac cycle. Formally, for a sequence of length $T$, we compute:

$$\mathcal{L}_{\text{EMD}} = \frac{1}{T}\big\|\text{cumsum}(\mathbf{p}) - \text{cumsum}(\mathbf{y})\big\|_1, \quad \mathbf{p} = (p_1, \ldots, p_T), \ \mathbf{y} = (y_1, \ldots, y_T). \quad (5)$$

This temporal EMD acts as a shape regularizer, ensuring that predicted event probabilities accumulate in a manner consistent with the true cardiac rhythm.

**Phase regression on $\mathbb{S}^1$:** Since cardiac phase is inherently periodic, we represent each target phase $\phi_t$ and its prediction $\hat{\phi}_t$ as unit vectors on the circle to avoid discontinuities at cycle boundaries and to ensure smooth transitions across beats. To enforce alignment between these circular embeddings, we regress directly on their direction using a cosine similarity loss:

$$\mathcal{L}_{\text{phase}} = \frac{1}{T}\sum_{t=1}^{T}\big(1 - \langle \hat{\phi}_t, \phi_t \rangle\big), \qquad \langle \mathbf{a}, \mathbf{b} \rangle = \frac{\mathbf{a}^\top \mathbf{b}}{\|\mathbf{a}\|_2 \|\mathbf{b}\|_2}. \quad (6)$$

This encourages predicted phases to point in the correct direction on $\mathbb{S}^1$, providing a smooth and periodicity-aware learning signal that remains robust to heart-rate variability.

**Total loss:** The full objective is a weighted sum of the different losses:

$$\mathcal{L}_{\text{total}} = \lambda_{\text{bce}}\mathcal{L}_{\text{BCE}} + \lambda_{\text{emd}}\mathcal{L}_{\text{EMD}} + \lambda_{\text{phase}}\mathcal{L}_{\text{phase}}, \quad (7)$$

with $\lambda$'s tuned so that component magnitudes are comparable at convergence.

### 2.5. Inference over long sequences

At test time, we apply a sliding temporal window with overlap and average overlapping probabilities per frame to obtain a single stabilized trace $\bar{p}_t \in [0, 1]$ over the full sequence. Peaks of $\bar{p}_t$ give ED frames, while $\hat{\phi}_t$ provides a continuous phase estimate.

### 3. Experiments

To assess the performance of our proposed method, we evaluate phase detection on both IVUS and CMR datasets.

### 3.1. Dataset and preprocessing

**IVUS:** We evaluate our method using 49 IVUS pullback sequences derived from the IM-PACT study, a serial multimodality imaging study performed at the Erasmus University Medical Centre (Hartman et al., 2021). Each sequence contains approximately 2,500 frames

Table 1: Comparison between different methods on the IVUS and CMR test datasets. Bajaj et al. (2021) denotes our reimplementation of their BiGRU model trained with their original setup. Huang et al. (2023) denotes our reimplementation of CARDIAN, a recent attention-based method for ED detection. "Ours (MSE loss)" uses our architecture but trained with Bajaj et al. (2021)'s MSE-based loss. "Ours (BCE only)" uses balanced BCE loss alone. "Ours (full)" corresponds to our complete multi-term loss (BCE+EMD+Phase).

| Method | IVUS | | | | CMR | | | |
|---|---|---|---|---|---|---|---|---|
| | AUROC ↑ | AD [s] ↓ | F1 ↑ | THM ↑ | AUROC ↑ | AD [s] ↓ | F1 ↑ | THM ↑ |
| Bajaj et al. (2021) | 0.84 | 0.17 | 0.86 | 0.73 | 0.92 | 0.09 | 0.43 | 0.40 |
| Huang et al. (2023) | 0.84 | **0.04** | 0.15 | 0.29 | 0.91 | **0.02** | 0.23 | 0.46 |
| Ours (MSE loss) | 0.79 | 0.16 | 0.88 | 0.75 | 0.50 | 0.10 | 0.41 | 0.37 |
| Ours (BCE only) | 0.88 | 0.07 | 0.98 | 0.91 | 0.95 | 0.03 | **0.78** | **0.76** |
| Ours (full) | **0.90** | 0.06 | **0.99** | **0.93** | **0.97** | **0.02** | **0.78** | **0.76** |

acquired at 16 fps with a pullback speed of 0.5 mm/s, yielding approximately 130,000 frames in total. IVUS data exhibit substantial inter-patient variability in vessel geometry, plaque morphology, catheter eccentricity, and imaging artifacts, providing considerable diversity in motion and appearance patterns. ED frame labels were derived by approximating frames located 6 frames before each R-peak in the corresponding ECG signal.

For training, we extract fixed-length temporal windows of $T \in \{64, 96, 128\}$ frames, sampled uniformly across each sequence. Unless stated otherwise, we use $T = 128$ for the main results. For each window we construct $C$ input channels corresponding to different *state encodings* $S \in \{S_0, S_1, S_{01}, S_{012}\}$, where

$$S_0 = \{x\}, \quad S_1 = \{\partial_t x\}, \quad S_{01} = \{x, \partial_t x\}, \quad S_{012} = \{x, \partial_t x, \partial_t^2 x\},$$

with temporal derivatives implemented as finite differences on the raw IVUS frames.

**CMR:** We use 535 short-axis cine MRI scans from patients with hypertrophic cardiomyopathy (HCM), a condition characterized by asymmetric left-ventricular thickening that challenges automated analysis (Vigneault et al., 2018). All CMR sequences were acquired at the Amsterdam University Medical Centre, and each study contained a stack of short-axis slices covering the full cardiac cycle with expert manual segmentations for the left ventricular cavity and the myocardium. We process CMR frames in native Cartesian coordinates to maintain end-to-end learning without segmentation or manual annotation dependencies. All scans are resampled to a common in-plane resolution and cropped to a fixed region of interest around the left ventricle using a fixed center-based cropping heuristic. Z-score normalization is subsequently applied to each individual scan.

For both IVUS and CMR datasets, data is split at the patient level into 70% training, 10% validation, and 20% test data.

### 3.2. Evaluation

We evaluate methods by assessing how well predicted end-diastolic (ED) frames align with the reference standard for both IVUS and CMR modalities. Furthermore, we compare our approach against established baselines and conduct systematic ablations to isolate the contribution of each component. Beyond aggregate metrics on the full test sets, we perform targeted analysis on artifact-heavy subsequences where appearance-based methods typically fail, revealing the practical benefits of our phase-aware formulation.

**Baselines and ablations:** We compare against the method of Bajaj et al. (2021) in two configurations: first, we reimplement their motion-signal-based BiGRU model trained with their original MSE loss, and second, we optimize our proposed architecture with an MSE loss to isolate the effect of our continuous phase representation from architectural differences. We additionally compare against CARDIAN (Huang et al., 2023), a recent attention-based approach for ED detection in IVUS. Since no official implementation was publicly available, CARDIAN was based on the published description and trained using identical data splits and evaluation protocols.

To disentangle the contribution of each component in our multi-objective formulation, we conduct systematic ablations examining six loss configurations: BCE alone, BCE combined with temporal EMD, and various combinations incorporating phase regression with different weighting factors $\lambda_{phase} \in \{0.1, 0.5, 1.0\}$. Additionally, we include a comparison to a Transformer-based temporal decoder, for which we keep the encoder, state encoding, loss configuration, optimizer settings, and window length fixed, and only replace the temporal decoder to isolate the effect of the backbone.

All models share identical data splits, state encodings, and optimization hyperparameters to ensure fair comparison across variants. Specifically, we train using the Adam optimizer with learning rate $10^{-3}$ and no learning rate scheduler, mini-batches of 8 windows, and early stopping on the validation loss. Unless stated otherwise, we use the $S_{01}$ state encoding. An ablation study comparing different state configurations is provided in Appendix B.

**Evaluation subsets:** Our evaluation encompasses two complementary levels of analysis. At the global level, we assess performance on the full test sets of both IVUS and CMR modalities. For targeted robustness analysis, we further identify representative IVUS pullback segments exhibiting pronounced artifacts from calcifications and catheter motion—specifically examining subsequences from four pullback cases. These challenging regions contain acoustic shadowing from calcified plaques, motion blur, and other common artifacts where accurate phase detection is limited due to limited visual cues in the imaging data.

**Evaluation metrics:** We employ a comprehensive set of metrics capturing both classification accuracy and temporal coherence. For frame-wise classification performance, we report area under the ROC curve (AUROC) and F1-score to measure the ability to correctly identify ED events. To quantify temporal precision across cardiac cycles, we compute mean absolute temporal distance (AD) in seconds between predicted and reference ED frames, and temporal harmonic mean (THM) which captures the consistency of inter-beat intervals. For challenging segments with severe artifacts, we additionally report prediction entropy (H) as a measure of model uncertainty. Predictions are first normalized to sum to one along the

Table 2: Ablation study of loss components and decoder architectures on the IVUS dataset. We evaluate six configurations combining different loss components. The phase column indicate the phase weighting factor $\lambda$. Metrics are the area under ROC curve (AUROC), mean absolute temporal distance (AD), and F1-score.

| Variant | Temporal decoder | BCE | EMD | Phase | AUROC ↑ | AD [s] ↓ | F1 ↑ |
|---------|------------------|-----|-----|-------|---------|----------|------|
| L1 | RNN | ✓ | – | – | 0.881 | 0.074 | 0.984 |
| L2 | RNN | ✓ | ✓ | – | 0.890 | 0.065 | 0.989 |
| L3 | RNN | ✓ | ✓ | 0.5 | **0.900** | **0.064** | 0.989 |
| L4 | RNN | ✓ | ✓ | 1.0 | 0.885 | 0.070 | **0.992** |
| L5 | RNN | ✓ | – | 0.1 | 0.876 | 0.079 | 0.981 |
| L6 | RNN | ✓ | – | 0.5 | 0.893 | 0.071 | 0.988 |
| T2 | Transformer | ✓ | ✓ | – | 0.669 | 0.142 | 0.948 |
| T3 | Transformer | ✓ | ✓ | 0.5 | 0.705 | 0.131 | 0.952 |

temporal dimension, after which entropy is computed as

$$H = -\sum_{t=1}^{T} p_t \log p_t,$$

where $p_t$ represents the normalized ED probability for frame $t$. Lower entropy values indicate more confident and stable predictions despite image degradation.

We note that AD is computed between predicted and reference ED events after matching within a tolerance window; therefore, low AD can occur even when recall is low. For this reason, we interpret AD jointly with THM. We note that AD is computed between predicted and reference ED events after matching within a tolerance window; therefore, low AD can occur even when recall is low. For this reason, we interpret AD jointly with THM.

**Computational efficiency:** Inference timing is evaluated on an NVIDIA A6000 GPU, on which our method achieves an end-to-end throughput of 66.2 frames/s, corresponding to approximately 4× real-time processing relative to a typical IVUS acquisition rate of 16fps. Model inference alone reaches 70.7 frames/s, with preprocessing adding 2.8s per sequence, of which most overhead originates from temporal gradient computation.

## 4. Results

### 4.1. Quantitative comparison

Table 1 presents the performance comparison between our proposed method and prior work on both IVUS and CMR test sets. For IVUS, the reimplementation of Bajaj et al. (2021) serves as our baseline, while variants of our architecture trained with different loss functions demonstrate the impact of our design choices. Our MSE-based model shows improved temporal metrics compared to the baseline. The BCE-only configuration further enhances performance across all metrics, while our full model incorporating BCE, EMD, and phase supervision achieves the best overall results with the highest AUROC and F1 scores,

Table 3: Quantitative comparison on challenging IVUS subsequences with severe artifacts. We report AUROC, AD, F1-score, and entropy (H) for our model without phase supervision (✗) and with phase supervision (✓). For models with phase supervision, $\lambda_{phase}$ is set to 0.5 in accordance with the best model identified from our ablation study (see Table 2).

| Sequence window | Phase | AUROC ↑ | AD [s] ↓ | F1 ↑ | H ↓ |
|---|---|---|---|---|---|
| Patient 1 (1100--1200) | ✗ | 0.672 | 0.196 | **0.824** | 0.450 |
| | ✓ | **0.907** | **0.073** | 0.706 | **0.275** |
| Patient 2 (700--900) | ✗ | 0.642 | 0.136 | 0.898 | 0.434 |
| | ✓ | **0.785** | **0.075** | **0.963** | **0.379** |
| Patient 3 (1600--1800) | ✗ | 0.734 | 0.097 | 0.986 | **0.271** |
| | ✓ | **0.788** | **0.070** | **1.000** | 0.337 |
| Patient 4 (1900--2100) | ✗ | 0.706 | 0.102 | 0.760 | **0.362** |
| | ✓ | **0.713** | **0.101** | **0.824** | 0.404 |

alongside the lowest temporal distance errors. CARDIAN (Huang et al., 2023) achieves low temporal distance (0.04s) but substantially lower F1 (0.15) compared to our method, indicating high precision but low recall: while the few detected ED events are temporally accurate, many ground-truth events remain undetected.

For CMR sequences, a similar pattern emerges where our BCE-based variants substantially outperform the MSE-based approaches. The baseline and MSE-based models exhibit lower discriminative ability despite reasonable temporal distances, suggesting difficulties in precise ED localization. In contrast, our BCE-only model shows marked improvements across all metrics, with our full model achieving the best performance, demonstrating the generalizability of our approach across imaging modalities. We refer to Appendix C for qualitative results on CMR phase estimation.

## 4.2. Impact of loss components and temporal backbones

Table 2 examines the contribution of individual loss components through systematic ablation on the IVUS validation set. The baseline BCE-only configuration (L1) provides strong performance, which is incrementally improved by adding temporal EMD regularization (L2). The inclusion of phase supervision with varying weights (L3-L6) maintains comparable aggregate metrics to BCE+EMD, with the combination of all three losses achieving optimal or near-optimal scores across metrics. Notably, phase supervision alone (L5, L6) can partially compensate for the absence of EMD regularization, though the complete formulation yields the most consistent results (L3).

Compared to the RNN decoder, the Transformer-based temporal decoder achieves lower performance across the metrics reported in Table 2. However, it follows the same ablation trend, i.e., adding the phase component improves the Transformer results relative to its no-phase variant.

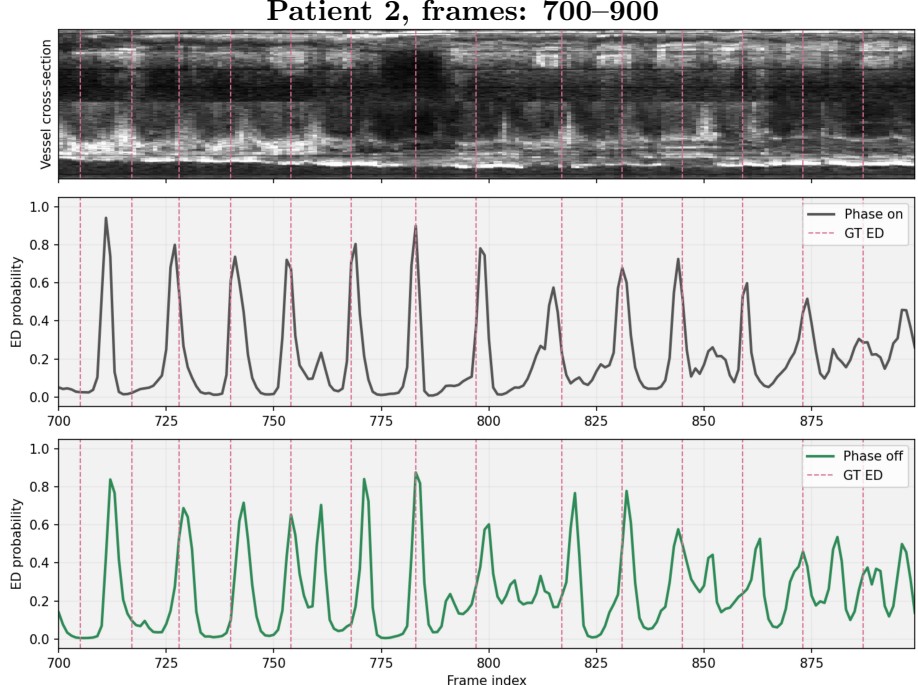

Figure 2: Qualitative comparison between our model without phase supervision (green) and our phase-aware model (grey) on a challenging IVUS subsequence with strong imaging artifacts. From top to bottom, we show the IVUS longitudinal cross-section over time over the selected frame range and the corresponding frame-wise ED probabilities; vertical dashed lines indicate the ground-truth ED frames.

### 4.3. Out-of-distribution performance

While aggregate metrics demonstrate strong overall performance, targeted analysis on challenging IVUS segments with severe imaging artifacts reveals the practical benefits of phase supervision. Table 3 and Figure 2 compare models with and without phase regularization on four representative pullback segments containing calcifications, acoustic shadowing, and catheter motion artifacts. Additional qualitative results for the remaining outlier cases are presented in Appendix D.

Across all examined segments, the phase-supervised model consistently outperforms the baseline without supervision. The improvements manifest most clearly in AUROC and AD metrics, with the phase-aware model maintaining more stable predictions as evidenced by lower entropy values in most cases. A qualitative comparison in Figure 2 illustrates this difference: without phase supervision, the model produces fragmented predictions with multiple spurious peaks within cardiac cycles, while the phase-supervised variant maintains single, well-localized peaks despite severe image degradation.

## 5. Discussion and conclusion

Our experiments demonstrate that directly modeling cardiac phase as a continuous variable on the unit circle yields robust phase detection that maintains accuracy even when image appearance deviates from the training distribution. The phase-supervised model consistently outperforms baselines on challenging IVUS segments containing calcifications and acoustic shadowing (Table 3), correctly identifying cardiac phase despite image degradation. This appearance-invariant performance is crucial for downstream tasks such as motion quantification and registration (van Herten et al., 2024), which require reliable phase alignment regardless of local image artifacts or pathological variations. Despite the moderate patient count, the substantial frame-level diversity and inter-patient variability in our IVUS dataset support the generalizability of these findings.

Beyond improving robustness, our framework eliminates the dependency on ECG gating that constrains current IVUS acquisition protocols. By learning phase directly from image sequences, we enable retrospective analysis of existing data and open possibilities for detecting pathological periodicities. Future work could leverage this periodicity-aware representation to identify non-standard cardiac rhythms or arrhythmias directly from the learned phase trajectories. Additionally, incorporating neural ordinary differential equations (Chen et al., 2018) could enforce even smoother phase evolution, explicitly modeling the continuous dynamics of cardiac motion rather than discrete frame transitions.

We additionally evaluated alternative temporal decoders, including a Transformer-based decoder (see Table 2). In our experiments, the RNN-based decoder consistently achieved the best performance, which we attribute to the inductive bias often expressed by the RNN architecture. The hidden-state update naturally enforces temporal continuity and can make the underlying dynamics easier to learn in our windowed, limited-data regime.

The key insight driving our performance gains is that mapping phase predictions onto $\mathbb{S}^1$ provides a geometrically consistent representation that naturally encodes cardiac periodicity. This circular embedding eliminates discontinuities at cycle boundaries and ensures that phase relationships remain well-defined under varying heart rates. Our results confirm that this principled treatment of phase as a cyclic variable, combined with multi-objective optimization balancing discrete event detection and continuous phase regression, establishes a robust foundation for standardized phase detection across cardiovascular imaging modalities.

## Acknowledgments

The project was partially funded by the call HORIZON-EIC-2022-PATHFINDERCHAL-LENGES-01 "CARDIOGENOMICS" from HORIZON European Innovation Council Grants / European Commission (DCM-NEXT project; project number 101115416) and partially funded by a private-public partnership grant provided by Health Holland, with contributions from B. Braun Melsungen AG, Germany, and Infraredx, Inc., Bedford, MA, USA (DEBuTLRP TKI-PPP, grant no. NCT04765956).

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

## Appendix A. Encoder architecture

**ResNet frame encoder:** For each subsequence, we extract a per-frame latent representation with a pseudo-3D ResNet encoder. The network applies pairs of $3 \times 3 \times 1$ convolutional layers with BatchNorm3D and LeakyReLU activations, followed by a residual skip connection. After each block, a stride, $(2, 2, 1)$, convolutional downsampling layer further reduces the spatial resolution while preserving the temporal dimension. Repeating this pattern over multiple stages yields a compact spatio-temporal feature map, which is subsequently aggregated into a low-dimensional latent representation that serves as input to the temporal classifier. The complete setup of the encoder is shown in figure 3.

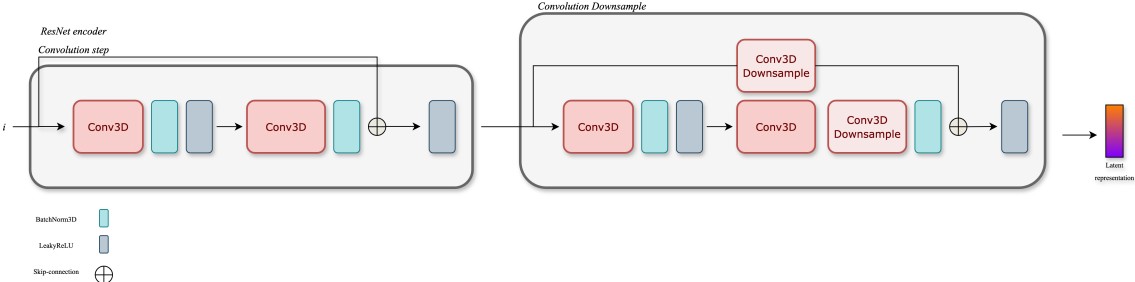

Figure 3: ResNet-style frame encoder used to obtain latent representations from IVUS subsequences. Each block consists of two Conv3D–BatchNorm3D–LeakyReLU layers with a residual skip connection (*Convolution step*), followed by a Conv3D downsampling block that reduces spatial resolution while keeping the temporal resolution fixed (*Convolution Downsample*). The final feature map is projected to a compact latent representation that is passed to the temporal phase detector.

## Appendix B. State encoding ablation

An ablation on state encodings constituting use of the $i^{th}$ temporal derivative for state $S_i$ is presented in Table 4. We use a temporal context window of 128 frames in combination with the L3 loss variant (see Table 2). Note that these ablations were conducted with minor differences in model architecture, and therefore results differ slightly from the best results in Table 2. Based on these results, $S_{01}$ was selected as the default configuration, as $S_{012}$ provided negligible additional performance gain while incurring higher computational cost.

## Appendix C. Qualitative results: CINE MRI

In Figure 4 we show three representative CINE MRI examples from the test set. For each patient, the top row displays selected short-axis frames with semi-transparent overlays of the left ventricle (LV) blood pool, myocardium, and right ventricle (RV). The bottom plot shows the left ventricle blood pool volume and the predicted ED probability over all frames, together with the ground-truth ED frame. These examples illustrate that the predicted ED

Table 4: Ablation study of state configurations on the IVUS dataset. Area under ROC curve (AUROC), mean absolute temporal distance (AD), and F1-score are presented.

| States | AUROC ↑ | AD [s] ↓ | F1 ↑ |
|--------|---------|----------|------|
| $S_0$ | 0.815 | 0.096 | 0.966 |
| $S_1$ | 0.884 | 0.069 | 0.988 |
| $S_{01}$ | 0.887 | 0.069 | **0.990** |
| $S_{012}$ | **0.894** | **0.067** | 0.987 |

probabilities follow the expected volume curves across a range of cardiac morphologies and image qualities.

## Appendix D. Qualitative results: IVUS

In Figure 5, we show additional examples from the IVUS test set for different patients.

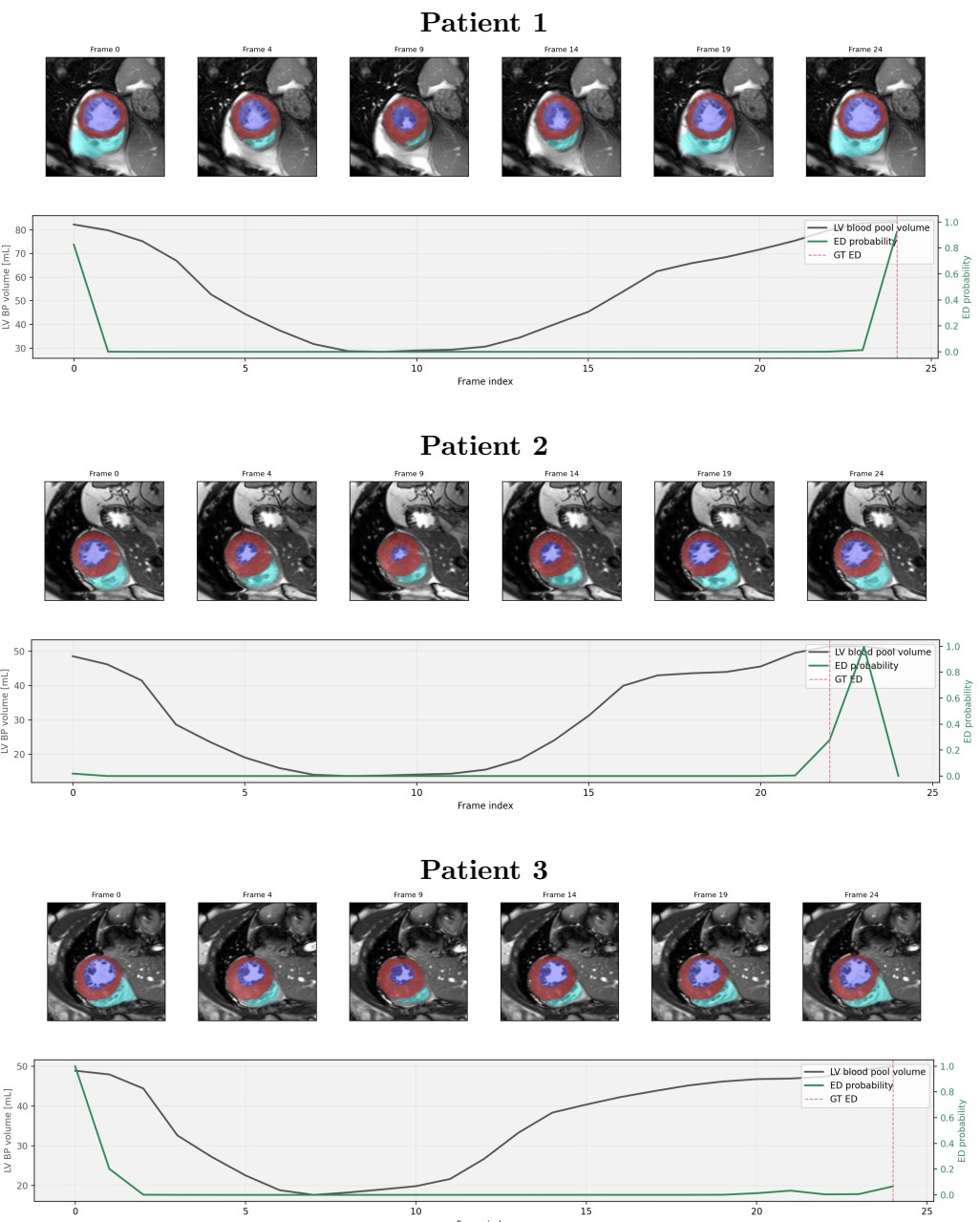

Figure 4: Additional qualitative CINE MRI examples from three test patients. For each patient, the top row shows selected short-axis frames with overlays of LV blood pool (blue), myocardium (red), and RV (cyan). The bottom plot displays LV blood pool volume and predicted ED probability as a function of frame index; vertical dashed lines indicate the ground-truth ED frame.

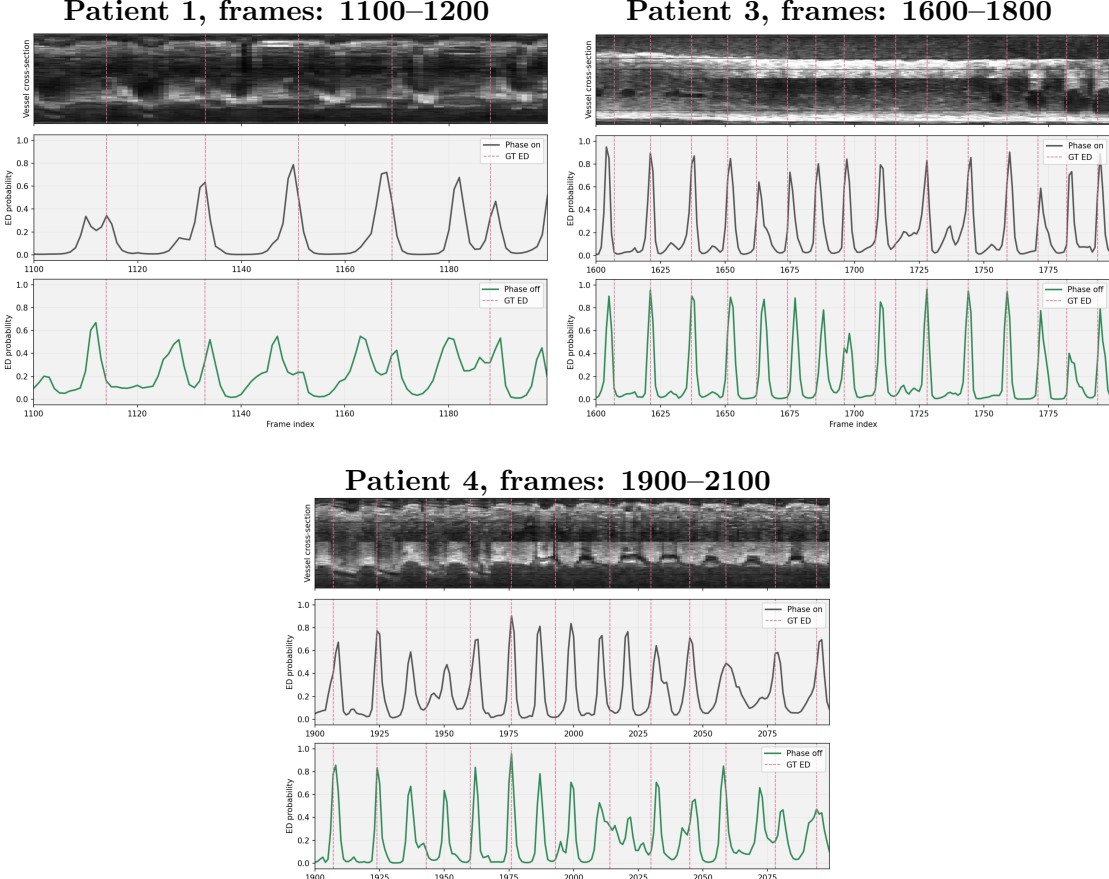

Figure 5: Additional qualitative examples on test-set patients with heavy artifacts. For each case we show the IVUS cross-section and predicted ED probabilities, following the layout of Figure 2 in the main text.

