# OpenReview forum: "CyclePhase: Robust phase detection in cardiovascular imaging through cyclic motion estimation"
_MIDL.io/2026/Conference — MIDL 2026 Poster_

### Official Review · Reviewer_fFvP · 2026-01-04

**Confidence:** 4
**Preliminary Rating:** 5

**Summary:**

The paper addresses the problem of cardiac phase detection in cardiovascular imaging, where existing methods typically treat phase estimation as discrete frame classification, making them sensitive to appearance variations and imaging artifacts. The authors propose a unified deep learning framework that models cardiac phase as a continuous cyclic variable, enabling temporally coherent phase estimation across entire cardiac cycles. The model consists of a framewise CNN encoder followed by a temporal RNN, jointly predicting end-diastolic events and a continuous phase representation embedded on the unit circle. The approach is evaluated on both intravascular ultrasound and cine MRI, including targeted analysis on artifact IVUS subsequence’s, within a framework that does not rely on segmentation masks.

Although no new architecture is proposed, the novelty lies in a principled reformulation of cardiac phase detection as continuous cyclic regression on the unit circle, supported by a multi-objective optimization strategy.

**Strengths:**

•	The topic and the solution proposed is very interesting and manuscript is well written and clearly presented, addressing a fundamental problem in cardiovascular imaging.

•	Conceptually novel formulation of phase detection as continuous cyclic regression.

•	Evaluated on both IVUS and cine MRI within a unified framework that avoids reliance on segmentation masks

•	The experimental section is well structured, including comparison to a relevant baseline and ablations, and targeted analysis on challenging, artifact-based IVUS cases.

**Weaknesses:**

The following points concern minor clarifications that would improve reproducibility and clarity:

• For clarity, it could be helpful to explicitly state that the “with phase supervision” results in Table 3 use the same phase loss weight as the full model selected in Table 2 (as suggested by the λ_phase ablation).

• Page 6: For CMR pre-processing, images are cropped to a fixed ROI around the left ventricle. It would be helpful to clarify whether this ROI is obtained using a fixed heuristic (e.g., a centre crop or predefined bounding box) or derived from available segmentations, to avoid any ambiguity regarding segmentation dependence.

• In Section 2.1 (Input transformations), it would improve reproducibility to briefly clarify how the centre c used for the IVUS polar transformation is determined in practice.

• Evaluation metrics section: It would be helpful to briefly clarify how entropy is computed, as this would improve reproducibility.

**Detailed Comments:**

• The provided GitHub link is currently not accessible.

**Justification Of The Preliminary Rating:**

The paper introduces a principled reformulation of cardiac phase detection as continuous cyclic regression, leading to improved temporal coherence compared to classification-based approaches. The method is well designed, carefully evaluated on IVUS and cine MRI, and supported by ablation studies and targeted analysis on challenging cases. The contribution is technically sound, clearly presented, and of high relevance to the MIDL community.

**Questions To Address In The Rebuttal:**

• Addressing comment in weakness section.

---

> ### Author Response · Authors · 2026-01-25
> **Response to Reviewer fFvP**
>
> We thank the reviewer for the positive evaluation and for the helpful suggestions aimed at improving clarity and reproducibility. We have addressed each point below.
>
> **(W1)** For clarity, it could be helpful to explicitly state that the “with phase supervision” results in Table 3 use the same phase loss weight as the full model selected in Table 2 (as suggested by the λ_phase ablation).
>
> **(A1)** We agree that this is a helpful clarification. We now explicitly state in the revised manuscript that the “with phase supervision” results reported in Table 3 use the same phase loss weight (λ_phase) as selected for the full model in Table 2, as determined by the λ_phase ablation study. The caption for **Table 3** now includes:\
> *“[...] For models with phase supervision, $\lambda_{phase}$ is set to 0.5 in accordance with the best model identified from our ablation study (see Table 2).”*
>
> **(W2)** Page 6: For CMR pre-processing, images are cropped to a fixed ROI around the left ventricle. It would be helpful to clarify whether this ROI is obtained using a fixed heuristic (e.g., a centre crop or predefined bounding box) or derived from available segmentations, to avoid any ambiguity regarding segmentation dependence.
>
> **(A2)** We have clarified the CMR pre-processing step in Section 3.1. Specifically, we now state that the region of interest around the left ventricle is obtained using a fixed heuristic (centre-based cropping) and does not rely on segmentation masks or manual annotations. This ensures that the method remains fully end-to-end and segmentation-independent. **Section 3.1 (CMR)** now reads:\
> *“We process CMR frames in native Cartesian coordinates to maintain end-to-end learning without segmentation or manual annotation dependencies. All scans are resampled to a common in-plane resolution and cropped to a fixed region of interest around the left ventricle using a fixed center-based cropping heuristic. Z-score normalization is subsequently applied to each individual scan.”*
>
> **(W3)** In Section 2.1 (Input transformations), it would improve reproducibility to briefly clarify how the centre c used for the IVUS polar transformation is determined in practice.
>
> **(A3)** We thank the reviewer for this suggestion. We have clarified the IVUS polar transformation in Section 2.1. Specifically, we now state that the catheter is inherently positioned at the image center due to the IVUS acquisition geometry, and therefore the polar transformation is performed using the image center as the reference point. **Section 2.1 (IVUS images)** now reads:\
> *“In IVUS imaging, the vessel wall exhibits approximately circular geometry around the catheter, which is inherently positioned at the image center due to the acquisition geometry. By transforming from Cartesian to polar coordinates, we convert the vessel's radial expansion and contraction into a simpler vertical displacement pattern that convolutional networks can easily capture. Specifically, we map Cartesian coordinates $\mathbf{u}=(x,y)$ to polar coordinates $\mathbf{v}=(r,\theta)$ using the image center as the reference point $\mathbf{c}$”*
>
> **(W4)** Evaluation metrics section: It would be helpful to briefly clarify how entropy is computed, as this would improve reproducibility.
>
> **(A4)** We have expanded the evaluation metrics section to explicitly define how entropy is computed, including the exact formulation and normalization used. **Section 3.2 (Evaluation metrics)** has been updated as follows:\
> *“Predictions are first normalized to sum to one along the temporal dimension, after which entropy is computed as $H = -\sum_{t=1}^{T} t_i \log t_i,$ where $t_i$ represents the normalized ED probability for frame $t$. Lower entropy values indicate more confident and stable predictions despite image degradation.”*
>
> **(DC1)** The provided GitHub link is currently not accessible.
>
> We indeed found that our repository was still set to private, which we have now updated to be publically available. Thank you for making us aware of this.

---

> > ### Comment · Reviewer_fFvP · 2026-01-29
> > **comment**
> >
> > Thank you for the detailed response. The authors have adequately addressed all of my comments and modified the manuscript, and I have no further concerns.

---

### Official Review · Reviewer_iKCF · 2026-01-07

**Confidence:** 4
**Preliminary Rating:** 2
**Final Rating:** 4

**Summary:**

The authors propose method for cardiac phase detection  by using gradient-based input transformations to isolate motion from static anatomy. Their method is robust to appearance variations, such as calcifications, in intravascular ultrasound. The evaluation shows that explicitly modelling cardiac periodicity yields more accurate and temporally coherent phase detection compared to classification-based approaches. The proposed method eliminates the need for modality-specific preprocessing or segmentation masks, providing an end-to-end solution for cardiac motion characterization.

**Strengths:**

- I really like how the paper is structured, it is well written
- Figure 1 is very nice to read
- Very nice ablation study on the different loss terms
- Very good quantitative and quantitative analysis

**Weaknesses:**

- Missing a detailed method figure, especially since a lot of equations are used, to combine the informations into one place
- The authors use only one baseline to compare three of their own methods.. That seems a little bit too low, there should be more baselines than just one (fairly old one as well) to compare their methods against

**Detailed Comments:**

- A little bit too much equations, where most of them are not even referenced anywhere/later used.. For instance Eqn. 4 is pretty clear from the text making the equation itself obsolete. It is better to use the remaining space for some concept figures and detailed figures on the method itself.
- Figure 2 is poorly placed, should be placed before the Discussion and Conclusion section

**Justification Of Final Rating:**

The authors addressed all of my comments in detail, and also added more baselines which increased the quality of the paper. Because of this, I raised my initial rating from a weak reject to a weak accept.

**Justification Of The Preliminary Rating:**

The manuscript lacks a clear, consolidated figure to explain the proposed method in detail, making the approach difficult to follow, and the evaluation is limited to a single outdated baseline, which weakens confidence in the claimed improvements.

**Questions To Address In The Rebuttal:**

1. Method clarity:
The authors should provide a detailed method figure summarizing all key components model components in one place to improve readability and understanding, rather than using 7 equations. Equations are generally fine, but some are not necessary and the space could be used for Figures.
2. Insufficient baselines: The authors should (as best as they can) including multiple, stronger or more recent baselines for a fair evaluation of their method rather than using only a single (relatively old) baseline.

---

> ### Author Response · Authors · 2026-01-25
> **Response to Reviewer iKCF (Part 1)**
>
> We thank the reviewer for the positive assessment of the manuscript and the constructive suggestions. We agree with the identified weaknesses and have addressed them as follows.
>
> **(1)** Missing a detailed method figure, especially since a lot of equations are used, to combine the informations into one place.
>
> **(A1)** We thank the reviewer for this suggestion and agree that a consolidated method figure would substantially improve clarity. Therefore, we have revised **Figure 1** to provide a comprehensive overview of the complete pipeline in a single schematic. The updated figure now explicitly illustrates: (1) the modality-specific preprocessing steps (polar transformation for IVUS, center cropping for CMR), (2) temporal gradient extraction yielding the different state encodings ($S_0$, $S_1$, and $S_2$), (3) the framewise convolutional encoder producing per-frame embeddings, (4) the temporal RNN module, and (5) the dual prediction heads outputting both ED logits and unit-circle phase vectors. This consolidated visualization directly links the mathematical formulations in Section 2 to their corresponding architectural components. The **caption for Figure 1** now reads:\
> *“Overview of the proposed cardiac phase detection framework. Top: A sequence of IVUS or ($\vee$) cine MRI frames is processed by a framewise convolutional encoder, producing per-frame embeddings $e_1, \dots, e_n$ that are passed to a temporal RNN. The RNN outputs both binary end-diastolic (ED) logits and continuous phase vectors on the unit circle, enabling joint ED frame detection and smooth phase estimation. Bottom left: Modality-specific preprocessing pipelines showing polar transformation for IVUS and center cropping for CMR, followed by temporal gradient extraction yielding state encodings $S_0, S_1, \text{and } S_2$. Bottom right: Example output showing predicted ED events and corresponding ED probability trace over time.”*
>
> **(2)** The authors use only one baseline to compare three of their own methods. That seems a little bit too low, there should be more baselines than just one (fairly old one as well) to compare their methods against.
>
> **(A2)** We agree that the original evaluation included few baselines. To strengthen the experimental section, we have made two additions: (1) a comparison against a more recent state-of-the-art method for cardiac event detection, and (2) an alternative temporal modeling backbone evaluated under identical training conditions.
>
> **Additional baseline method.** We have added CARDIAN (Huang et al., 2023), a recent attention-based method for ED detection in IVUS. Since no official implementation was publicly available, we reimplemented CARDIAN based on the published description and trained it using identical data splits and evaluation protocols. The results in **Table 1** show that CARDIAN achieves low AD (0.04s on IVUS, 0.02s on CMR) but substantially lower F1 (0.15 on IVUS, 0.23 on CMR) compared to our method, indicating high precision but low recall: while the few detected ED events are temporally accurate, many ground-truth events remain undetected. To prevent misinterpretation of AD in isolation, we now explicitly state in **Section 3.2** that AD should be interpreted jointly with detection metrics. We describe this additional result in **Section 4.1**:\
> *"CARDIAN (Huang et al., 2023) achieves low temporal distance (0.04s) but substantially lower F1 (0.15) compared to our method, indicating high precision but low recall: while the few detected ED events are temporally accurate, many ground-truth events remain undetected."*
>
> **Alternative temporal backbone.** We have extended our ablation study by comparing the RNN temporal decoder with a Transformer-based temporal decoder, while keeping the frame encoder, state encoding, loss configuration, optimizer settings, and window length fixed to ensure a fair comparison. These results are reported in **Table 2** (rows T2 and T3). The RNN-based decoder consistently achieved better performance, which we attribute to its inductive bias toward temporal continuity, a property well-suited to cardiac motion signals in our windowed, limited-data regime. This comparison is discussed in **Section 5**:\
> *"We additionally evaluated alternative temporal decoders, including a Transformer-based decoder (see Table 2). In our experiments, the RNN-based decoder consistently achieved the best performance, which we attribute to the inductive bias often expressed by the RNN architecture. The hidden-state update naturally enforces temporal continuity and can make the underlying dynamics easier to learn in our windowed, limited-data regime."*
>
> These additions provide a more comprehensive evaluation of the proposed approach relative to existing methods.

---

> ### Author Response · Authors · 2026-01-25
> **Response to Reviewer iKCF (Part 2)**
>
> **(DC1)** A little bit too much equations, where most of them are not even referenced anywhere/later used. For instance Eqn. 4 is pretty clear from the text making the equation itself obsolete. It is better to use the remaining space for some concept figures and detailed figures on the method itself.
>
> **(A1)**  We thank the reviewer for this suggestion and understand the concern. Since the revised submission still fits within the available space, we kept the equations in the main text for completeness and to maintain a self-contained description. However, if the reviewer feels that this disrupts readability, we are happy to move the less essential equations to an appendix in the camera-ready version (during discussion period), while keeping only the key equations in the main paper.
>
> **(DC2)** Figure 2 is poorly placed, should be placed before the Discussion and Conclusion section
>
> **(A2)** We agree and have moved Figure 2 to a more appropriate location earlier in the manuscript, placing it directly before the Discussion and Conclusion section (now located in Section 4.3), so that it is introduced and interpreted in the corresponding section.

---

### Official Review · Reviewer_q3Yu · 2026-01-08

**Confidence:** 5
**Preliminary Rating:** 2
**Final Rating:** 4

**Summary:**

The authors propose to utilize motion-focused preprocessing, circular phase representation, and multi-objective optimization for phase detection in cardiovascular imaging. Their proposed model architecture follows an encoder-decoder architecture, where a framewise ResNet encoder is followed by a temporal RNN decoder.

**Strengths:**

(1) The authors utilized multiple datasets to evaluate their methods.

(2) The authors employ several loss functions in their framework and conduct an ablation study to assess the contribution of each loss component.

**Weaknesses:**

(1) The authors should consider comparing their RNN‑based temporal module with state‑of‑the‑art temporal modeling approaches, such as Transformers and TCNs.

(2) Building on the previous point, the authors only compare their method with one baseline method published in 2021. The authors should compare their method with more recent state-of-the-art methods.

(3) Although the authors incorporate motion‑focused preprocessing, its impact is not evaluated. Ablation studies are needed to justify the necessity and effectiveness of these preprocessing steps.

(4) As the dataset size is small, cross-validation should be conducted.

(5) Please add inference speed analysis for the proposed method.

**Detailed Comments:**

For (2), here are some references that might be helpful:

Huang, Xingru, et al. "CARDIAN: a novel computational approach for real-time end-diastolic frame detection in intravascular ultrasound using bidirectional attention networks." Frontiers in Cardiovascular Medicine 10 (2023): 1250800.

Balaji, G. N., and R. Venkatesan. "Optimized pyramidal convolution shuffle attention neural network based fetal cardiac cycle detection from echocardiograms." Biomedical Signal Processing and Control 98 (2024): 106701.

Li, Yuanshu, et al. "A deep learning based approach for automatic cardiac events identification." Biomedical Signal Processing and Control 100 (2025): 107164.

Patel, Sahaj, et al. "A Novel Framework for End-Diastolic and End-Systolic Frame Localization in Contrast and Non-Contrast Echocardiography Without Manual Annotations." Circulation 152.Suppl_3 (2025): A4373195-A4373195.

**Justification Of Final Rating:**

I would like to thank the authors for their response. The authors have conducted a major revision of the paper. The newly added ablation study, the additional experiments comparing against state‑of‑the‑art approaches, and the inference‑time analysis significantly improve the overall quality of the work. Based on these improvements, I am raising my rating to a weak accept.

**Justification Of The Preliminary Rating:**

While the application addressed in the paper is interesting, the work suffers from a lack of comparison with state‑of‑the‑art methods, and the evaluation section is overall weak. Given these limitations, I have to recommend a weak reject.

**Questions To Address In The Rebuttal:**

Please ensure that all identified weaknesses are thoroughly addressed, with particular attention to weaknesses (1), (2), (3), and (4).

---

> ### Author Response · Authors · 2026-01-25
> **Response to Reviewer q3Yu (Part 1)**
>
> We thank the Reviewer for their critical feedback, which we have addressed to the best of our ability.
>
> **(W1)** The authors should consider comparing their RNN‑based temporal module with state‑of‑the‑art temporal modeling approaches, such as Transformers and TCNs.
>
> **(A1)** We thank the Reviewer for this suggestion and agree that comparing alternative temporal backbones is important. To directly address this, we have extended our ablation study by comparing the RNN temporal decoder with a Transformer-based temporal decoder, while keeping the frame encoder, state encoding, loss configuration, optimizer settings, window length, training protocol, and evaluation setup identical to ensure a fair comparison. These new results are included in **Table 2 (rows T2 and T3)** of the updated manuscript, and described in Section 3.2.
>
> To make the experimental control explicit, we add the following sentence in **Section 3.2 (Baselines and ablations)**:\
> *“Additionally, we include a comparison to a Transformer-based temporal decoder, for which we keep the encoder, state encoding, loss configuration, optimizer settings, and window length fixed, and only replace the temporal decoder to isolate the effect of the backbone.”*
>
> We also report the observed trend in **Section 4.2 (Impact of loss components)**:\
> *“Compared to the RNN decoder, the Transformer-based temporal decoder achieves lower performance across the metrics reported in Table 2. However, it follows the same ablation trend: adding the phase component improves the Transformer results relative to its no-phase variant.”*
>
> Transformers model temporal dependencies through self-attention and can, in principle, capture long-range interactions across the full window. However, this flexibility comes with fewer built-in constraints on temporal smoothness and often requires more data to reliably learn temporal dynamics. RNNs, by design, propagate information through a recurrent hidden state, which provides a strong sequential inductive bias toward continuity and gradual evolution, properties that align well with cardiac motion signals. We hypothesize that, given our dataset size and fixed windowed training setup, this inductive bias helps the RNN learn the relevant temporal dynamics more efficiently, while the Transformer’s higher capacity may not translate into improved generalization under the same training budget. We have added a short discussion to contextualize this result and hypothesize why RNNs may be advantageous in our regime in **Section 5 (paragraph 3)**:\
> *“We additionally evaluated alternative temporal decoders, including a Transformer-based decoder (see Table 2). In our experiments, the RNN-based decoder consistently achieved the best performance, which we attribute to the inductive bias often expressed by the RNN architecture. The hidden-state update naturally enforces temporal continuity and can make the underlying dynamics easier to learn in our windowed, limited-data regime.”*

---

> ### Author Response · Authors · 2026-01-25
> **Response to Reviewer q3Yu (Part 2)**
>
> **(W2)** Building on the previous point, the authors only compare their method with one baseline method published in 2021. The authors should compare their method with more recent state-of-the-art methods.
>
> **(A2)** We thank the Reviewer for this suggestion and agree that additional comparisons strengthen the evaluation. As per one of the Reviewers’ suggested papers, we have added CARDIAN (Huang et al., 2023), a recent attention-based method for ED detection in IVUS that represents the closest recent work to our setting. Since no official implementation was publicly available, we reimplemented CARDIAN based on the published description and trained it using identical data splits and evaluation protocols.
>
> The results in Table 1 show that CARDIAN achieves low AD (0.04s on IVUS, 0.02s on CMR) but substantially lower F1 (0.15 on IVUS, 0.23 on CMR) compared to our method, indicating high precision but low recall. The method detects few ED events, but those it does detect are temporally accurate. This pattern highlights a limitation of evaluating AD in isolation. To prevent misinterpretation, we now explicitly state in Section 3.2 that AD should be interpreted jointly with detection metrics. We have made the following changes to the manuscript:
>
> The caption for **Table 1** now includes:\
> *"Huang et al. (2023) denotes our reimplementation of CARDIAN, a recent attention-based method for ED detection."*
>
> **Section 3.2 (Baselines and ablations)** now reads:\
> *"We compare against the method of Bajaj et al. (2021) in two configurations: first, we reimplement their motion-signal-based BiGRU model trained with their original MSE loss, and second, we optimize our proposed architecture with an MSE loss to isolate the effect of our continuous phase representation from architectural differences. We additionally compare against CARDIAN (Huang et al., 2023), a recent attention-based approach for ED detection in IVUS. Since no official implementation was publicly available, CARDIAN was based on the published description and trained using identical data splits and evaluation protocols."*
>
> **Section 4.1 (Quantitative comparison)** now includes:\
> *"CARDIAN (Huang et al., 2023) achieves low temporal distance (0.04s) but substantially lower F1 (0.15) compared to our method, indicating high precision but low recall: while the few detected ED events are temporally accurate, many ground-truth events remain undetected."*
>
> **Section 3.2 (Evaluation metrics)** now clarifies:\
> *"We note that AD is computed between predicted and reference ED events after matching within a tolerance window; therefore, low AD can occur even when recall is low. For this reason, we interpret AD jointly with THM."*
>
> **(W3)** Although the authors incorporate motion‑focused preprocessing, its impact is not evaluated. Ablation studies are needed to justify the necessity and effectiveness of these preprocessing steps.
>
> **(A3)** We thank the reviewer for this suggestion, and agree that an ablation on state encodings provides valuable insights. Such an ablation was conducted at an earlier stage of our work, which we now include in the updated manuscript. In these experiments, we identified $S_{01}$ as an optimal trade-off between performance and efficiency, where only including $S_0$ resulted in significantly lower accuracy, while $S_{012}$ incurred more overhead for negligible performance gains. We include these results in **Appendix A** of the updated manuscript, with the newly added **Table 4**:\
> *“An ablation on state encodings constituting use of the $i^{th}$ temporal derivative for state $S_i$ is presented in Table 4. We use a temporal context window of 128 frames in combination with the L3 loss variant (see Table 2). Note that these ablations were conducted with minor differences in model architecture, and therefore results differ slightly from the best results in Table 2. Based on these results, $S_{01}$ was selected as the default configuration, as $S_{012}$ provided negligible additional performance gain while incurring higher computational cost.”*
>
> We refer to this newly included appendix at the end of **Section 3.2 (Baselines and ablations)**:\
> *“An ablation study comparing different state configurations is provided in Appendix A.”*

---

> ### Author Response · Authors · 2026-01-25
> **Response to Reviewer q3Yu (Part 3)**
>
> **(W4)** As the dataset size is small, cross-validation should be conducted.
>
> **(A4)** We thank the reviewer for raising this point and appreciate the opportunity to clarify. While the patient count may appear modest, each IVUS pullback sequence comprises approximately 2,500 frames, yielding 127147 labeled frames across training, validation, and test sets. Furthermore, IVUS data exhibit substantial inter-patient variability due to differences in vessel geometry, plaque morphology, catheter eccentricity, motion patterns, and imaging artifacts, providing considerable diversity in the motion and appearance patterns encountered during training. This is further supported by our qualitative analysis, which demonstrates robust generalization to challenging and atypical cases. We now clarify this in **Section 3.1 (IVUS)**:\
> *“[...], yielding approximately 130,000 frames in total. IVUS data exhibit substantial inter-patient variability in vessel geometry, plaque morphology, catheter eccentricity, and imaging artifacts, providing considerable diversity in motion and appearance patterns.”*
>
> We further include a note on the generalization of our method in the **Discussion (paragraph 1)**:\
> *“Despite the moderate patient count, the substantial frame-level diversity and inter-patient variability in our IVUS dataset support the generalizability of these findings.”*
>
> Nonetheless, we agree that cross-validation would further strengthen our conclusions. Due to time and resource constraints in this rebuttal period, we opted to prioritize the inclusion of an additional baseline and a transformer-based temporal decoder. However, we are prepared to conduct 3-fold cross-validation during the discussion period should the reviewer consider this essential for acceptance.
>
> **(W5)** Please add inference speed analysis for the proposed method.
>
> **(A5)** We thank the reviewer for this suggestion. We have added an inference-time analysis in Section 3.2 of the updated manuscript, reporting timing metrics on an NVIDIA A6000 GPU.
> Our method achieves an end-to-end throughput of 66.2 frames/s (excluding disk I/O), with model inference alone reaching 70.7 frames/s. Given that IVUS sequences are typically acquired at 16fps, this corresponds to approximately 4$\times$ real-time processing speed, demonstrating suitability for practical deployment. Preprocessing, including gradient computation, adds modest overhead (2.8s per sequence, of which 71% is attributable to temporal gradient computation), while data transfer costs are negligible (<150ms per sequence). We include these results in **Section 3.2 (Computational efficiency)** of the updated manuscript:\
> *"Inference timing is evaluated on an NVIDIA A6000 GPU, on which our method achieves an end-to-end throughput of 66.2 frames/s, corresponding to approximately 4$\times$ real-time processing relative to a typical IVUS acquisition rate of 16fps. Model inference alone reaches 70.7 frames/s, with preprocessing adding 2.8s per sequence, of which most overhead originates from temporal gradient computation."*

---

### Author Rebuttal · Authors · 2026-01-25

**Rebuttal:**

We would like to thank all reviewers for their constructive feedback that has helped us improve our manuscript. The most prominent changes to the manuscript address concerns raised by multiple reviewers: we have added CARDIAN (Huang et al., 2023) as an additional baseline comparison (Table 1), and included a Transformer-based temporal decoder in our ablation study (Table 2, rows T2 and T3). We have also revised Figure 1 to provide a more comprehensive pipeline overview and added several clarifications to improve reproducibility. We include detailed responses to the individual Reviewers in their respective responses, and look forward to answering any remaining concerns in the discussion period.

**Supporting Material:**

/attachment/7a8a6afbe40b5a759b38977fb62301bdb5d74dbb.pdf

---

### Meta-Review · Area_Chair_s94R · 2026-02-03

**Recommendation:** Accept (Poster)
**Confidence:** 5

**Metareview:**

This work proposes a method for cardiac phase detection in cardiovascular imaging. The originality of the approach lies in modeling the cardiac phase as a continuous cyclic variable, enabling temporally coherent phase estimation across the entire cardiac cycle. The approach is evaluated on both intravascular ultrasound and cine MRI data, including a targeted analysis of IVUS subsequences containing imaging artifacts. The evaluation shows that explicitly modeling cardiac periodicity enables more accurate and temporally coherent phase detection than classification-based approaches.

The discussion and rebuttal phases were handled seriously by the authors, which significantly improved the manuscript overall and led to a consensus among the reviewers regarding the quality of the final version. For these reasons, I have decided to accept this article.

---

### Decision · Program_Chairs · 2026-02-13

Accept (Poster)